# Therapeutic Options in Hydatid Hepatic Cyst Surgery: A Retrospective Analysis of Three Surgical Approaches

**DOI:** 10.3390/diagnostics14131399

**Published:** 2024-06-30

**Authors:** Alin Mihetiu, Dan Bratu, Bogdan Neamtu, Dan Sabau, Alexandra Sandu

**Affiliations:** 1County Clinical Emergency Hospital of Sibiu, 550245 Sibiu, Romania; alin_mihetiu@yahoo.com (A.M.); bogdan.neamtu@ulbsibiu.ro (B.N.); prof_dansabau@yahoo.com (D.S.); alexandrasandu96@yahoo.ro (A.S.); 2Faculty of Medicine, Lucian Blaga University of Sibiu, 550169 Sibiu, Romania

**Keywords:** hydatid hepatic cyst, open surgery, laparoscopy, special surgical devices

## Abstract

Hydatid disease is endemic in certain geographical areas where animal breeding is common, frequently challenging the medical services in these regions. Hydatid cysts most often affect the liver, with damage to other organs accounting for around one-third of the total cases. The alternative to interventional or pharmacological approaches is surgical treatment, available in variants such as laparoscopy, laparoscopy with special instruments for hydatid disease, or open surgery. This article aims to analyze the outcomes of these three types of surgical approaches, considering preoperative indications, operative techniques and efficiency, and immediate and long-term postoperative results. A total of 149 patients from two different surgical units were analyzed over a period of seven years. It was observed that males were more affected by this pathology (53.02%), with the majority of patients coming from rural areas (62.42%). The distribution by surgical procedure type showed that 50.34% were operated on using open surgery, 33.56% by means of a laparoscopic approach with the usual instruments, and 16.11% by means of a laparoscopic approach with special instruments. The laparoscopic procedure with special instruments presented a lower rate of conversion to open surgery compared to the usual laparoscopic approach (*p* = 0.014). The analysis of the average operative duration revealed statistically significant differences between the three types of surgical techniques (*p* < 0.05), noting that interventions with specialized instruments had the shortest duration, while open surgery had the longest operative time (72.5 ± 27.23 min vs. 154 ± 52.04 min). In terms of intraoperative complications, they were documented in 8.34% of cases for the group operated on with special instruments, in 12.24% of cases for the standard laparoscopy group, and in 16% of cases for the open surgery group. Maximal cystectomy was the preferred method for resolving these cysts using minimally invasive surgery (*p* < 0.001), while Lagrot pericystectomy was preferred in the open approach (*p* < 0.001). The most frequent postoperative complication was biliary fistula (24.16%), encountered in varying percentages across each technique but without significant statistical difference (*p* > 0.05). Open surgery was associated with a longer length of hospitalization compared to minimally invasive procedures (*p* < 0.05), a higher number of late postoperative complications (*p* = 0.002), and a significantly higher number of recurrences (*p* < 0.001) compared to the other two techniques. The present study highlights the effectiveness of minimally invasive surgery for hydatid cysts as a safe alternative with fewer complications and superior results compared to open surgery. Additionally, it provides a comparative analysis of these surgical approaches (special instruments, standard laparoscopy, and open surgery) to hydatid disease for the first time. Under the circumstances where pharmacological treatment is recommended as a supportive measure before and after procedures, and using medication alone as the primary treatment option shows only modest efficacy, there is a necessity to consider invasive treatment methods. Percutaneous procedures represent the least invasive form of treatment, yielding results comparable to surgery in terms of efficacy. However, their effectiveness is influenced by factors such as the cyst’s stage of development, its location, and the challenges in achieving complete intra-procedural isolation. Laparoscopy, particularly when using specialized instruments tailored to the tactical and technical demands of managing hydatid disease, serves to address the limitations of percutaneous methods. Open surgery’s role is increasingly restricted, primarily serving as a fallback option in laparoscopic procedures or in cases complicated by hydatid disease. In conclusion, despite the rising popularity of percutaneous methods, surgery remains a viable therapeutic option for treating hydatid disease. Minimally invasive surgical interventions are increasingly versatile and yield comparable outcomes, further solidifying the role of surgery in its management.

## 1. Introduction

Hydatid disease is a zoonosis caused by the Echinococcus tapeworm, most commonly Echinococcus granulosus and Echinococcus multilocularis. The annual incidence of hydatid cysts ranges from 1 to 200 per 100,000 inhabitants. The disease is endemic in geographical areas where animal breeding is prevalent, such as the Mediterranean region, the Middle East, Africa, New Zealand, Australia, and South America. However, due to global mobility, the disease can now be detected worldwide [1,2].

After passing through the digestive and portal systems, the Echinococcus tapeworm primarily inoculates the liver in 70% of cases, the lungs in 25% of cases, and other organs in 5% of cases. It can simultaneously affect the liver and lungs in 5–13% of cases, with the right lobe of the liver being affected in 65% of hepatic cases [3].

Historically, hydatid disease was known about in Ancient Egypt, described in the Ebers Papyrus and later in the Corpus Hippocraticum, with contributions from Celsus and Galen [4]. The first real attempts to treat this disease began in the late 19th century, involving the puncturing and evacuation of cystic contents, but these maneuvers often led to relapses and infections. Improvements in antiparasitic treatment, asepsis conditions, and the introduction of antibiotics have significantly improved surgical outcomes. Open surgery was the only treatment method for nearly a century, evolving from simple punctures to partial excisions and ideal cystectomies. The advent of laparoscopy and the PAIR (Puncture, Aspiration, Instillation, Reaspiration) technique three decades ago introduced minimally invasive options for treating this condition [5].

Given the potential for systemic evolution and complications in about one-third of patients, early diagnosis and effective therapeutic solutions are crucial. Imaging modalities like ultrasound, CT, and MRI allow the early diagnosis and assessment of disease progression, localization, and relationships with surrounding anatomical structures, facilitating an appropriate therapeutic plan [6]. The integration of artificial intelligence in imaging, enabling 3D or 4D imaging and differential diagnosis predictions (e.g., distinguishing from biliary mucinous cystic neoplasm), marks a significant advancement in the diagnosis of hydatid disease [7,8].

Pharmacological treatment, while effective, is most useful as a neoadjuvant or adjuvant to invasive procedures. Medications typically include benzimidazole group drugs, especially albendazole, administered alone or with praziquantel. In abdominal hydatidosis, nitazoxanide is also necessary, showing superior results when combined with standard therapy compared to monotherapy or double therapy (albendazole and praziquantel) [9,10].

Exclusive pharmacological treatment can reduce hydatid scolices but is less effective for large cysts or multiple hydatid cysts. Additionally, pharmacological therapy’s duration, cost, and side effects pose limitations [11].

Cystic content inactivation involves instilling scolicidal agents to destroy active parasitic elements post-aspiration, reducing intraoperative contamination risk. Historically, various substances have been used as scolicidal agents, but many have been abandoned due to adverse reactions or inefficacy [12]. Currently, alcohol solutions and hypertonic serum are used for their optimal scolicidal effects and reduced adverse reactions. Promising scolicidal effects have been observed in vitro with new substances and nanoparticles, though these studies are still in preclinical phases [13,14]. Promising results have been reported in the use of radiofrequency and microwave ablation as treatment methods in the hepatic locations of hydatid cysts, but the methods are used on a small scale and require additional studies on statistically significant groups of patients [15].

PAIR is the least invasive method but has a high recurrence rate and risks complications such as contamination. It is not widely accessible and is limited by the number, content, and location of cysts, being almost exclusively used for hepatic cysts and contraindicated in pulmonary or cerebral locations [5,16].

Surgical treatment can be performed via traditional open surgery, laparoscopy, or laparoscopy with specially designed instruments for hydatid surgery. While numerous comparative studies between open procedures and laparoscopy exist, an analysis including all three surgical approaches has not yet been conducted.

This study aims to perform an analytical comparison of these surgical procedures from the perspectives of preoperative, intraoperative, and postoperative data. It will examine the impact of preoperative clinical, imaging, and demographic data on the choice of a specific procedure. The intraoperative effectiveness of each procedure will be assessed in terms of the operative duration, intra-procedural complications, and conversion rates. Additionally, the study will focus on evaluating the postoperative outcomes for each technique.

## 2. Materials and Methods

The included cases comprised patients diagnosed with hydatid cysts located in the liver and extrahepatic abdominal regions. The study included exclusively adult patients who underwent surgical intervention. Exclusion criteria were patients without a surgical indication, those who underwent the PAIR procedure, those who refused surgical intervention, and pediatric patients.

All patients underwent preoperative imaging investigations, either with computed tomography (CT) or abdominal MRI. Elective cases received a preoperative preparation with 400 mg of albendazole twice daily for 28 days. Patients weighing under 60 kg received albendazole at a dosage of 5 mg/kg of body weight daily in two divided doses. This pharmacological treatment continued for an additional month postoperatively.

The surgical methods analyzed in this study included the open surgery approach, the standard laparoscopic approach, and the laparoscopic approach using special instruments specifically designed for hydatid disease surgery.

### 2.1. Open Surgery

The open surgical approach was performed via a Kocher-type right subcostal laparotomy, with extension to a left subcostal incision if intraoperative conditions necessitated, or through a median laparotomy. The surgical field was isolated with compresses soaked in a scolicidal agent (either hypertonic serum or alcohol). This was followed by the puncturing and evacuation of the cystic contents, aspiration, and inactivation with scolicidal solutions, and subsequent reaspiration of the inactivated content (Figure 1).

The approach to the remaining cyst wall and cavity varied between conservative procedures (operculectomy, pericystectomy) and radical procedures (cystectomy).

### 2.2. Laparoscopic Approach

Pneumoperitoneum was achieved through a supraumbilical approach using a Veress needle. The laparoscopic approach was performed using 10 mm and 5 mm trocars, depending on the location of the hydatid cyst. The supramesocolic area was isolated using alcohol-soaked mesh or mesh impregnated with hypertonic serum. After identification, the cyst was punctured, and the cystic content was aspirated. This was followed by the inactivation of the cyst with scolicidal substances and subsequent reaspiration. The cyst could be simply drained, or laparoscopic operculectomy, pericystectomy, or cystectomy could be performed (Figure 2 and Figure 3).

### 2.3. Laparoscopic Approach with Specialized Instruments

In addition to the standard laparoscopic instrumentation, the procedure involves the use of a transparent coaxial rig with anchoring hooks at the level of the cystic wall, a vacuum cleaner with a double instillation and suction mechanism, and a device for fragmenting the cystic content when it is not efficiently suctioned due to the intracystic material (proligera membrane, vesicle daughters).

The instrumentation for hydatid cysts (State Office for Inventions and Trademarks Patent No. 120809/30.04.2008) represents an adaptation of standard laparoscopic instrumentation to the intraoperative needs related to the characteristics of the cyst wall and contents (Figure 4 and Figure 5).

This technique allows all suction, inactivation, and reaspiration maneuvers to be carried out under direct vision, minimizing the risk of spillage. The operative time is shortened by streamlining the extraction and insertion maneuvers of the instruments according to the operative stage, providing psychological comfort to the surgeon due to the low risk of contamination.

## 3. Results

### 3.1. Demographics Data

The average age of the patients was 50.49 ± 16.58 years, with the gender distribution being 53.02% (n = 79) for males and 46.98% (n = 7) for females.

The predominant environment of origin was a rural one (62.42%), with 76.51% of the patients presenting at an appointment and 23.49% as an emergency (Table 1).

A significant percentage of patients who presented as emergencies had complications from hydatid disease (62.86%). In comparison, only 20.18% of elective patients had complications, a difference that was statistically significant (Chi^2^ = 23.14, df = 1, *p* < 0.001). Regarding the number of cysts, 65.56% of cases had a single hepatic localization, while 34.44% had multiple localizations.

The analysis of presentation modes based on their origin environment reveals that 32.26% of urgent cases originated from rural areas, whereas only 8.93% came from urban areas, a statistically significant finding (*p* < 0.001).

A comparative examination of demographic data related to each surgical approach is summarized in Table 1. It shows significant correlations between age and the chosen procedure, as well as the history of surgical interventions for hydatid disease. This underscores a tendency towards open surgery following unsuccessful percutaneous or prior surgical procedures.

### 3.2. Operative Data

In terms of the surgical approach, 50.34% of the cases were managed with open surgery, 33.56% with a standard laparoscopic approach, and 16.11% with a laparoscopic technique using specialized instruments. Overall, minimally invasive approaches accounted for 49.66% of cases, with 32.43% utilizing instruments specifically designed for hydatid surgery.

The specialized instruments for hydatid cyst surgery facilitated a more frequent approach to multiple cysts compared to standard laparoscopy (41.67% vs. 34%).

The distribution of echinococcal infection was predominantly in the right lobe of the liver (67.79%), with single locations being more frequently found in the entire right liver (68.32%).

When analyzing the type of approach in cases with complications of hydatid disease, open surgery was used for 36% of patients, standard laparoscopy was used for 26% of patients, and laparoscopy with special instruments was used for 20.83% of patients. However, no significant statistical differences were found when comparing these three approaches (*p* > 0.05).

The conversion rate was lower for interventions using specialized instruments (4.17%) compared to the standard laparoscopic approach (24%), with the following statistical results: U = 515.5, z = −2.69, asymptotic *p* = 0.007, exact *p* = 0.014, r = 0.22.

Regarding the duration of the surgical intervention, minimally invasive laparoscopic procedures were shorter than open laparotomy approaches, with the shortest operative times observed in interventions using specialized instruments (Table 2, Figure 6 and Figure 7).

### 3.3. Postoperative Data

Intraoperative complications occurred in 20 patients (13.51%), with the most common being intraoperative hemorrhage and spillage of cystic contents. In the group using specialized instruments, 8.34% experienced complications, with diaphragmatic laceration being the most frequent (4.17%). In the standard laparoscopy group, 12.24% experienced complications, with cystic content spillage occurring in 8.16% of cases. The open surgery group had a 16% complication rate, with hemorrhage being the most common complication (8%).

Maximal cystectomy, which involves a near-total cystectomy, was the predominant approach in surgeries using specialized instruments, accounting for 48% of these procedures. This technique was used in 44% of standard laparoscopic approaches and only 8% of open surgeries, with significant statistical differences among the three types of approaches (*p* < 0.001).

A total of 63.16% (*p* = 0.026) of maximal cystectomies were performed using open surgery, while the remaining 36.84% were performed laparoscopically. Lagrot-type pericystectomy was performed in 97.78% of open surgeries (*p* < 0.001) and in 2.22% of laparoscopic procedures. Simple pericystectomies were mostly performed laparoscopically, accounting for 85.71% (*p* < 0.01), with the remaining 14.29% performed via laparotomy.

Additional surgical maneuvers were necessary in 15.44% of cases, mainly cholecystectomy, T-tube biliary drainage, and diaphragmatic suture. The distribution of these maneuvers according to the type of surgical approach is summarized in Table 3.

In the immediate postoperative period, complications were recorded in 26.85% of patients, with the most frequent being postoperative biliary fistula, occurring in 36 patients (24.15%). This complication was more prevalent in open surgical interventions, affecting 12.75% (n = 19) of these patients. However, no statistically significant relationship was found between the type of surgical approach and the occurrence of immediate complications (Chi^2^ = 11.01, df = 10, *p* = 0.35) (Table 4).

Following surgery, postoperative management included ERCP in 12.75% of patients to address prolonged biliary fistulas. Additional procedures included laparoscopic drainage for subdiaphragmatic hematoma, reintervention for cystic cavity superinfection, and thoracentesis for pleurisy. The analysis of biliary leakage duration showed no statistically significant differences among surgical approaches: 3.89 ± 9.03 days for open surgery, 2.92 ± 5.61 days for the laparoscopic approach, and 1.17 ± 2.84 days for specialized instrument use (*p* > 0.05). A total of 21 patients (14.09%) experienced complications unrelated to the surgical procedure. Among these, the most common complication was Clostridium difficile infection, accounting for 42.85% of cases. Of these, 33.33% occurred post open surgery (*p* = 0.037).

Two deaths were recorded: one following open surgery and another followed a standard laparoscopic approach, resulting in an overall study mortality rate of 1.34%. Specifically, the mortality rates were 1.35% in the minimally invasive surgery group and 1.33% in the open surgery group, with no statistically significant difference observed (*p* = 0.4) (Table 5).

Patients undergoing open surgery had an average hospitalization length of 16.32 ± 9.22 days, compared to 11.24 ± 5.68 days for standard laparoscopy and 6.71 ± 2.16 days for specialized instrument use. Comparative analysis of these variables is presented in Table 6 and Figure 8.

Complications occurring after discharge, termed as distant complications, were observed in 14.76% (n = 22) of cases, with the most prevalent being parietal defects such as subcostal and median incisional hernias in 8.32% of cases, all of which were operated on via an open approach (*p* = 0.002). Additionally, compared to the overall hydatid recurrence rate of 4.03%, a rate of 3.36% was noted with the classic surgical approach (*p* < 0.001).

## 4. Discussion

In regions with endemic Echinococcus infections, the human manifestation of this disease poses a significant public health challenge. Despite its seemingly benign nature, this condition exhibits expansive, metastatic, and recurrent attributes reminiscent of neoplastic diseases. The multifaceted nature of hydatid disease contributes to elevated treatment costs, stemming from the necessity of pre- and post-operative pharmacotherapy, intricate imaging diagnostics, and complications associated with its localization in vital organs such as the liver and lungs. Furthermore, heightened mortality and recurrence rates, estimated by the WHO at 2.2% and 6.5%, respectively, underscore the gravity of this condition and the imperative for comprehensive management strategies.

The management of hydatid cysts is complex, being pharmacological, percutaneous, and surgical, and requires a multidisciplinary collaboration between an infectious disease physician, an interventional radiologist, a gastroenterologist, and a surgeon.

The WHO recommendations in the therapy of hepatic hydatid cysts are oriented according to the modified WHO Gharbi classification and cyst sizes, ranging from pharmacological monotherapy to combined pharmacological therapy—PAIR/surgery to liver transplantation in severe liver parenchymal damage [17,18].

For CE1 hydatid cysts measuring under 5 cm, treatment with albendazole alone or in combination with praziquantel is recommended. However, it is important to note that the cure rate with single pharmacological therapy is only 18%. This underscores the rationale for combining drug therapy with invasive procedures or considering drug therapy alone in cases where percutaneous or surgical interventions are contraindicated. These considerations highlight the need for a comprehensive and individualized approach to treatment, taking into account the specific characteristics and circumstances of each case [19,20].

Studies reporting positive outcomes with the sole use of drug therapy often stem from small patient cohorts, case series, or individual case reports. Some of these reports advocate for the prolonged administration of drug therapy. However, it is essential to interpret these findings cautiously, considering the limitations inherent in studies of this nature, such as small sample sizes and potential biases. Further research, ideally through well-designed clinical trials with larger populations, is needed to elucidate the efficacy and safety of drug therapy as a standalone treatment for hydatid cysts comprehensively [21].

Given the absence of large-scale population studies demonstrating unequivocal efficacy, we do not advocate for monotherapy with imidazoles alone. Instead, such therapy may be considered as an initial treatment for CE1 stage infections, subject to subsequent review based on imaging and immunological assessments [22].

The percutaneous procedures such as PAIR (Puncture, Aspiration, Injection, Reaspiration), MoCaT (Modified Catheterization Technique) and PAID (Puncture Aspiration Instillation Drainage) have limited applicability, usually to CE1 and CE3a types, smaller than 5 cm, preferably with a single location that is technically and imaging-accessible [5,10,23,24].

The widespread adoption of these procedures is thoroughly justified by their minimally invasive nature, offering both technical and anesthetic advantages such as local anesthesia or sedation, reduced hospital stays, and expedited patient recovery. PAIR is particularly indicated for patients who either decline surgical intervention, have significant contraindications for surgery, experience treatment failure with drug therapy, or encounter hydatid recurrences post-surgery. However, it is crucial to note that PAIR is contraindicated in cases where cysts have fistulized into the bile ducts or in pulmonary locations of the hydatid cyst. Additionally, performing the procedure in abdominal sites beyond the liver, such as the intestines or mesentery, poses technical challenges due to the heightened risk of cyst rupture and involvement of adjacent organs.

Recurrence in PAIR techniques or those derived from PAIR ranges between 3% and 20%, seemingly more common in the original PAIR technique compared to MoCaT or PAID/PAIDS. Major complications associated with these techniques occur in 16.4% of patients, while minor complications appear in 21.9% of cases [25,26,27].

When comparing these findings with the results of our study, similarities emerge in both the number of complications and the recurrence rate. Minimally invasive surgical procedures exhibited fewer complications compared to PAIR techniques; however, surgical interventions entail higher costs due to the type of anesthesia and longer hospital stays required [16,28].

This aspect warrants discussion because the indications for surgery in hydatid disease are broader compared to those for PAIR. Surgery is typically reserved for advanced stages, challenging locations, and cases complicated by hydatid disease, thus often involving patients with a higher complexity index.

Zaharie et al., in a comparative study between laparoscopic and open surgical approaches, underscore the superiority of laparoscopy in terms of surgical duration, intraoperative and postoperative complications, as well as hospital stay duration [29]. This observation is corroborated by several other studies, yielding results akin to those found in our investigation [30,31,32,33,34].

The main criticisms of the laparoscopic approach at the beginning of its implementation were the increased risk of spillage of hydatid contents, difficulties in addressing challenging locations, extrahepatic locations, and limitations regarding radical interventions. Over time, laparoscopy has addressed these so-called weaknesses. Laparoscopic surgery, particularly due to preoperative and intraoperative imaging, has become a safe and efficient method, superior to open surgery concerning radical procedures such as total cystectomy, unregulated and regulated hepatic resections, segmentectomies, and even laparoscopic lobectomies.

Radical procedures are applicable in laparoscopic surgery, showing comparable results to conservative methods, with some studies associating a lower rate of complications such as biliary fistula and recurrence [35,36,37].

Multiple extrahepatic locations or abdominal hydatidosis were once firmly indicated for open surgery. However, the laparoscopic approach can also be successfully used in these locations, being a safe method without the risk of intraoperative spillage and offering superior benefits compared to an extensive laparotomy. As a solution to the risk of intraoperative spillage during laparoscopic interventions, various techniques have been devised to allow surgical stages to be performed in a safe and efficient environment [38,39,40].

In the present study, the analysis of a specialized surgical approach for hydatid disease compared to standard laparoscopy and open surgery demonstrates its superiority in various aspects, including operation duration, intraoperative and postoperative complications, and length of hospitalization. However, the applicability of minimally invasive surgery to cases with complications of hydatid disease is limited, particularly in instances of rupture of hydatid contents into the bile ducts, often necessitating external biliary drainage through open surgical intervention.

The integration of ERCP (endoscopic retrograde cholangiopancreatography) into the treatment of hydatid disease offers an alternative approach after bile duct clearance, thereby reducing the risk of biliary leakage. Moreover, in cases complicated by postoperative biliary fistula, ERCP application significantly diminishes the amount and duration of drainage compared to conservative management [41].

Upon reviewing the three surgical procedures, it is evident that minimally invasive surgery had a shorter operative duration and fewer intraoperative and postoperative complications compared to the conventional open approach. In regions where hydatid cyst cases are common and laparoscopic expertise is limited, the duration of laparoscopic interventions may exceed that of open surgery. The specialized instrument approach designed for hydatid cysts offers a safe surgical environment, minimizing contamination risks and necessitating fewer maneuvers for instrument handling, thus reducing surgical duration compared to conventional laparoscopy.

Postoperative complications related to the surgical approach, while more prevalent in open surgery, did not exhibit statistically significant differences compared to minimally invasive surgery. Similar results were observed in the analysis of distant recurrences. The extended hospitalization duration, notably higher in open surgery, is linked to prolonged antibiotic therapy and prolonged exposure to the hospital environment, elevating the risk of non-surgical infections such as Clostridium difficile enterocolitis. This finding was further supported by our analysis [42].

In open surgery, the most prevalent distant postoperative complication often manifests as post-incisional hernias. The anatomical dynamics, including oblique or transverse sections through muscle-aponeurotic structures, vascular networks, and nerve pathways, along with the distribution of abdominal pressure forces, render parietal defects more likely, particularly in comparison to median incisions. Furthermore, the use of foreign materials in parietal sutures during open surgery can lead to local inflammatory and granulomatous reactions over time, increasing the likelihood of incisional hernia development—a scenario seldom encountered in minimally invasive surgery [43,44,45,46].

Minimally invasive surgery for hydatid cysts expands its range of indications and performances using intraoperative imaging (intraoperative ultrasound) or artificial intelligence models on preoperative imaging segments [7].

The combination of indocyanine green fluorescence imaging with laparoscopy achieves safer resection with fewer sacrifices of liver parenchyma and simultaneously reduces the risk of bile duct or vascular injuries [47].

The progress of robotic surgery has enabled hydatid cysts to be approached using this method, with results similar to laparoscopic surgery. However, more comprehensive studies are needed to conduct an accurate analysis regarding the benefit-to-cost efficiency ratio [48].

## 5. Conclusions

Surgical management remains a cornerstone in addressing abdominal hydatid disease, particularly in hepatic involvement, where percutaneous modalities may be constrained.

Assessing three distinct surgical approaches for hepatic hydatid disease underscores the superiority of minimally invasive methods. These techniques offer expanded indications, superior intraoperative precision, and improved postoperative outcomes. The utilization of specialized instruments facilitates more precise cyst management, resulting in enhanced surgical and recovery outcomes compared to both traditional open surgery and standard laparoscopy.

## Figures and Tables

**Figure 1 diagnostics-14-01399-f001:**
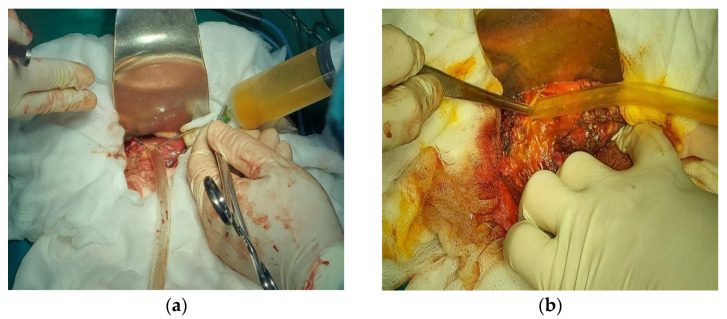
(**a**) Aspiration of hydatid content; (**b**) evacuation of contents after scolicide instillation.

**Figure 2 diagnostics-14-01399-f002:**
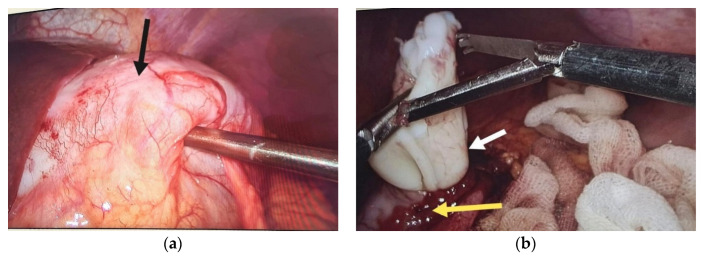
(**a**) Hydatid cyst (black arrow); (**b**) cystic wall (yellow arrow) and proligera membrane (white arrow).

**Figure 3 diagnostics-14-01399-f003:**
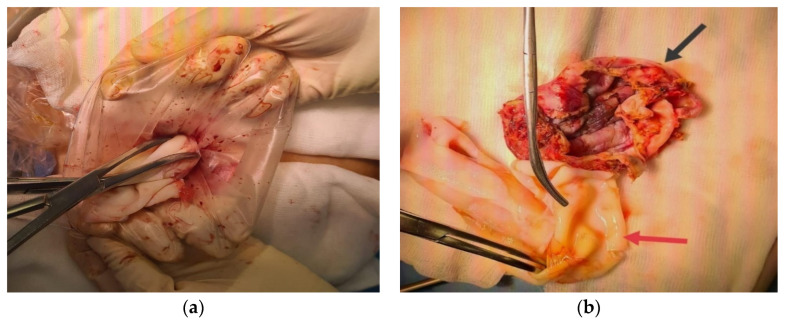
(**a**) Cyst extraction using an endobag; (**b**) cyst wall specimen (black arrow), proligera membrane (red arrow).

**Figure 4 diagnostics-14-01399-f004:**
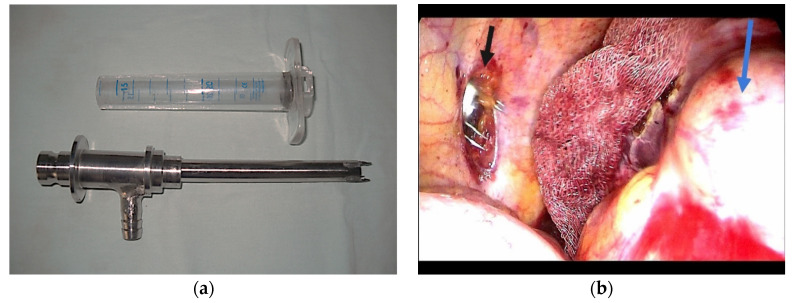
(**a**) Special device for hydatid cyst; (**b**) trocar with anchoring hooks (black arrow) and cyst wall (blue arrow).

**Figure 5 diagnostics-14-01399-f005:**
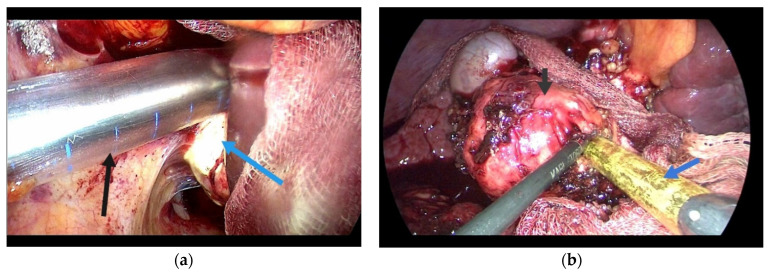
(**a**) Coaxial trocar (black arrow) anchored to the cyst (blue arrow); (**b**) evacuation of the hydatid cyst content (black arrow) through the trocar (blue arrow).

**Figure 6 diagnostics-14-01399-f006:**
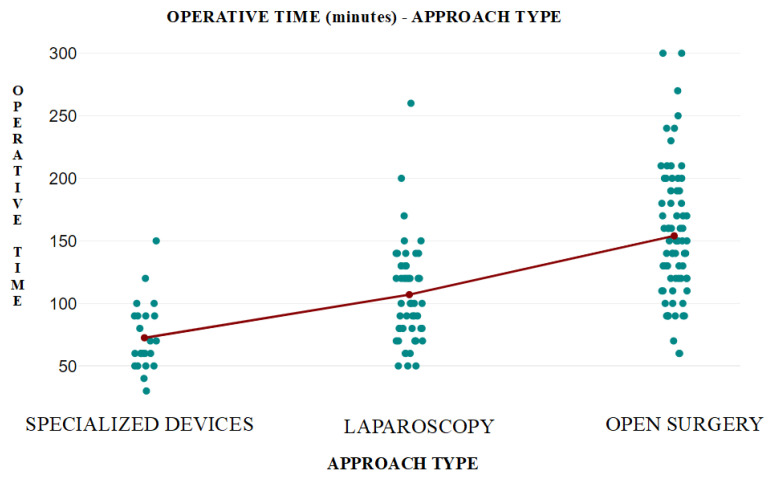
Graphic representation of the relationship between the type of surgical approach and operative time.

**Figure 7 diagnostics-14-01399-f007:**
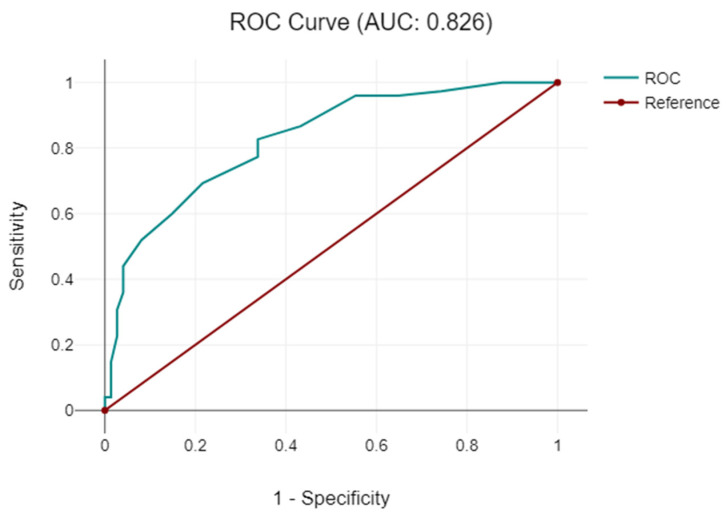
Receiver operating characteristic curve (ROC curve) with the analysis between the open surgery approach and the duration of the surgical intervention.

**Figure 8 diagnostics-14-01399-f008:**
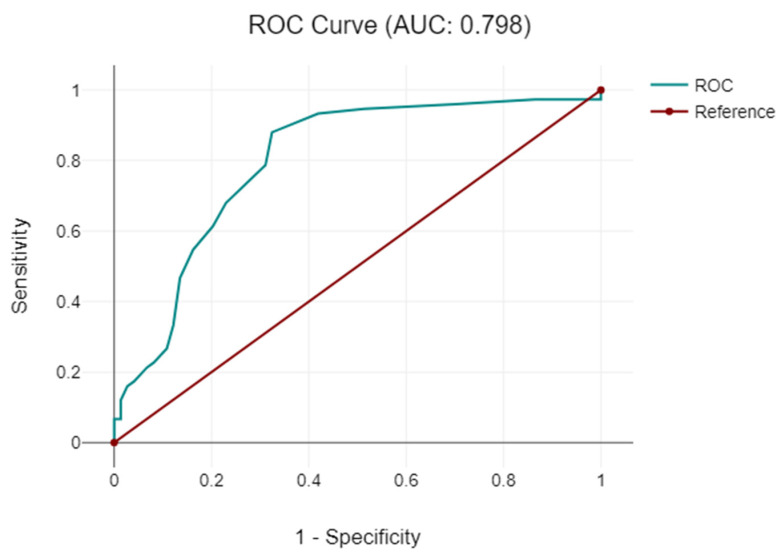
Receiver operating characteristic curve (ROC curve) with the analysis between the open surgery approach and hospitalization duration.

**Table 1 diagnostics-14-01399-t001:** The distribution according to surgical approach and demographic variables.

Variable		Specialized Instruments	*p* Value	Laparoscopy	*p* Value	Open Surgery	*p* Value
Age (years)		54.83 ± 14.33	0.045	53.38 ± 15.67	0.045	47.17 ± 17.33	0.045
Gender	Male	70.83%	0.052	52%	0.859	48%	0.216
	Female	29.17%	0.47	48%	0.47	52%	0.47
Area	Urban	41.67%	0.359	44%	0.359	32%	0.359
	Rural	58.33%	0.253	56%	0.359	68%	0.359
Presentation type	Elective	87.5%	0.142	76%	0.917	73.33%	0.356
	Emergency	12.5%	0.142	24%	0.917	26.67%	0.356
Number of cysts	Singular	58.33%	0.407	66%	0.96	68%	0.56
	Multiple	41.67%	0.407	34%	0.96	32%	0.56
Previous history for hydatid disease		8.33%	0.252	2%	0.09	12%	0.018

**Table 2 diagnostics-14-01399-t002:** The duration of operative time in relation with the type of surgical approach.

SURGICAL TECHNIQUE		N	%	Mean	Minimum	Maximum	Mean ± Std.	Statistical Results
Operative time (minutes)	LAPAROTOMY	75	50.34%	154	60	300	154 ± 52.04	z = 5.61, *p* < 0.001, OR =1.03 95% CI = 1.02–1.04
	LAPAROSCOPY	50	33.56%	107	50	260	107 ± 40.01	z = 2.8, *p* = 0.005, OR = 0.99, 95% CI = 0.98–1
	SPECIAL DEVICES	24	16.11%	72.5	30	150	72.5 ± 27.23	z = 4.58, *p* < 0.001, OR = 0.96, 95% CI = 0.94–0.97

**Table 3 diagnostics-14-01399-t003:** Surgical procedures associated with basic intervention.

APPROACH TYPE						
SPECIALIZED DEVICES		LAPAROSCOPY		OPEN SURGERY		
INTERVENTION TYPE	% IN APPROACH TYPE	*p* Value	% IN APPROACH TYPE	*p* Value	% IN APPROACH TYPE	*p* Value
LAPAROSCOPIC CHOLECYSTECTOMY	0.67%	*p* = 0.023	2.68%	*p* = 0.023	-	-
DIAPHRAGMATIC SUTURE	0.67%	*p* = 0.32	0%	-	0.67%	*p* = 0.32
T TUBE DRAINAGE	0%	-	0.67%	*p* = 0.153	3.36%	*p* = 0.015
SPLENECTOMY	0%	-	0%	-	0.67%	*p* = 0.5
CYSTO-JEJUNAL ANASTOMOSIS	0%	-	0%	-	0.67%	*p* = 0.5
OPEN CHOLECYSTECTOMY	0%	-	0%	-	4.03%	*p* = 0.014
OMENTECTOMY	0%	-	0%	-	0.67%	*p* = 0.5
ERCP	0%	-	0.67%	*p* = 0.33	0%	-

**Table 4 diagnostics-14-01399-t004:** Incidence of postoperative complications by surgical technique.

IMMEDIATE POSTOPERATIVE COMPLICATIONS	SPECIALIZED DEVICES		LAPAROSCOPY		OPEN SURGERY		Total	
-	n	%	n	%	n	%	n	%
NO COMPLICATIONS	19	12.75%	35	23.49%	55	36.91%	109	73.15%
BILLIARY LEAKEAGE	4	2.68%	13	8.72%	19	12.75%	36	24.16%
EMPHIZEMA	1	0.67%	0	0%	0	0%	1	0.67%
HEMATOMA	0	0%	1	0.67%	0	0%	1	0.67%
ILEUS	0	0%	1	0.67%	0	0%	1	0.67%
GASTRIC FISTULA	0	0%	0	0%	1	0.67%	1	0.67%
Total	24	16.11%	50	33.56%	75	50.34%	149	100%

**Table 5 diagnostics-14-01399-t005:** Postoperative outcomes in relation to the surgical procedure.

Variable	Specialized Instruments	*p* Value	Laparoscopy	*p* Value	Open Surgery	*p* Value
Immediate postoperative complications	20.83%	0.458	30%	0.539	26.67%	0.96
Suplimentary interventions during hospital stay	12.5%	0.804	18%	0.338	12%	0.459
Biliary drainage period (days)	1.17 ± 2.84	0.175	2.92 ± 5.61	0.541	3.89 ± 9.03	0.198
Other complications	0%	-	8.16%	0.178	24%	0.003
Unfavorable outcome	0%	-	2%	0.63	1.33%	0.99
Long term complications	0%	-	8%	<0.001	22.67%	<0.001

**Table 6 diagnostics-14-01399-t006:** Statistical analysis of hospitalization duration.

Variables	Mean Difference	t	*p*
Specialized devices—Laparoscopy	−4.53	−2.47	0.015
Specialized devices—Open surgery	−9.61	−5.54	<0.001
Laparoscopy—Open surgery	−5.08	−3.76	<0.001

## Data Availability

Data are contained within the article.

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
