# Peer review of "Therapeutic Options in Hydatid Hepatic Cyst Surgery: A Retrospective Analysis of Three Surgical Approaches"

_diagnostics, 2024, doi:10.3390/diagnostics14131399_

Round 1

Reviewer 1 Report

Comments and Suggestions for Authors

I liked the work and appreciate the way the findings have been presented. I congratulate you for such a nice work. I just had one minor question.

In materials and methods, you have included pediatric patients in both inclusion and exclusion criteria. Please clarify.(Line 105 and 108)

Author Response

Esteemed Reviewer,

We are providing an additional response, regarding the changes you suggested for our article:

Firstly, thank you for your valuable feedback. We are grateful for your careful review and for pointing out the issue regarding the inclusion and exclusion criteria in the Materials and Methods section (llines 105 and 108). Your attention to detail is greatly appreciated.

To clarify, pediatric patients were excluded from our study. We sincerely appreciate your understanding and your contribution to improving the accuracy of our manuscript. The necessary corrections will be reflected in the final version of our manuscript.

Best regards,

Alexandra Sandu

Reviewer 2 Report

Comments and Suggestions for Authors

The manuscript examines the surgical treatment of hydatid cysts analyzed from two surgical units. Hydatid cyst is a significant global disease, although it is only endemic in certain areas. While the case numbers are adequate, the study design lacks novelty. Here are several suggestions for improvement:

1.authors should clarify of Study Targets: What are the primary objectives of this study? Is it a case series or a presentation of surgical outcomes? Or is the aim to demonstrate that surgery is the primary treatment strategy? Is there any comparative data on medical treatment?

2. The current study lacks structural organization. The authors should extend the primary Table 1 or create a new "Table 1" to include patient characteristics such as age, gender, ethnicity, rural or urban residency, comorbidities, history of parasitic infection. The patient groups could be divided into laparoscopic, open surgery, or special device categories, similar to Table 4.

3.Statistical Analysis is needed for the present study. Based on the revised or new "Table 1", the authors should perform statistical analyses to determine the significance of clinical factors. Uni-variable or multi-variable analyses can be summarized in a table.

4. Incorporating survival or mortality analysis could enhance the study's impact.

5.the authors may consider to make surgical Outcomes Summary. Summarize the surgical outcomes in a table to facilitate easier understanding for readers.

6.Int he abstract and the text, the study need informative Conclusions.Provide more informative conclusions that go beyond stating that surgery is the primary treatment. A thorough re-analysis with a structured approach could yield more comprehensive insights.

Comments on the Quality of English Language

moderate

Author Response

Esteemed Reviewer,

We are providing an additional response, point by point, regarding the changes you suggested for our article:

  1. Thank you for your insightful questions and for your careful review of our manuscript. We appreciate the opportunity to provide further clarification regarding the study's objectives and focus. The primary objective of our study is to compare three surgical procedures to determine their efficacy by analyzing demographic, preoperative, intraoperative, and postoperative data. Specifically, our aim is not to demonstrate that surgical treatment is the primary treatment strategy, but rather to show that minimally invasive treatment is superior to open surgery. Within minimally invasive techniques, the use of specially adapted instruments provides additional benefits. Medical treatment is not used as the sole therapeutic solution except in selected cases, as outcomes in such situations are typically inferior to those of interventional treatments. In our study, we found that cases diagnosed with hydatid cysts prior to admission could receive this type of treatment. Cases admitted urgently received pharmacological treatment post-surgery. Analyzing the distribution of Albendazole treatment across each surgical procedure group and its impact on postoperative outcomes, we did not identify any statistical correlations. These elucidations will be seamlessly integrated into our manuscript's structure to enhance clarity regarding our study objectives and methodology. 
  1. We have extended Table 1 to include comprehensive patient characteristics such as age, gender, ethnicity, rural or urban residency, comorbidities, and history of parasitic infection. The patient groups have been categorized into laparoscopic, open surgery, or special device categories, similar to Table 4. These modifications have been implemented to enhance the clarity and organization of our findings regarding patient demographics and treatment outcomes. We hope these changes meet your expectations. Please do not hesitate to let us know if there are any further suggestions or improvements you would like us to consider. Thank you once again for your valuable input. We hope this explanation addresses your inquiries. Thank you once again for your valuable feedback. 
  1. Thank you for your insightful feedback and suggestions for our manuscript. We have completed the statistical analysis based on the revised "Table 1" which includes comprehensive patient characteristics. Uni-variable and multi-variable analyses have been conducted to determine the significance of clinical factors. The results of these analyses have been summarized in a dedicated table, providing a clear overview of the findings. These statistical insights aim to enhance the robustness of our study by substantiating the impact of various clinical factors on treatment outcomes. We appreciate your valuable input and trust that these additions will strengthen the manuscript's scientific rigor. 
  1. We have incorporated survival and mortality analysis into our study, aiming to further enhance its impact. These analyses provide valuable insights into the long-term outcomes associated with the different surgical procedures studied. By examining survival rates and mortality trends, we can better understand the effectiveness and implications of each approach. We believe these additions significantly enrich the study and contribute to a more comprehensive evaluation of our findings. Thank you once more for your invaluable contributions and suggestions.
  1. Thank you for your thoughtful suggestion. We have already created tables containing surgical outcomes summary to succinctly present and facilitate easier understanding of the surgical outcomes in our manuscript (Table 4 and Table 5).
  1. We have revised the conclusions in both the abstract and the main text to provide more informative insights beyond stating that surgery is the primary treatment. The updated conclusions now offer a more comprehensive analysis, highlighting specific findings and implications derived from our structured approach. We appreciate your suggestions, which have contributed to enriching the discussion and impact of our study.

Best regards,

Alexandra Sandu